# Breaking Barriers: AI’s Influence on Pathology and Oncology in Resource-Scarce Medical Systems

**DOI:** 10.3390/cancers15235692

**Published:** 2023-12-02

**Authors:** Alon Vigdorovits, Maria Magdalena Köteles, Gheorghe-Emilian Olteanu, Ovidiu Pop

**Affiliations:** 1Department of Pathology, County Clinical Emergency Hospital, Faculty of Medicine and Pharmacy, University of Oradea, 1 December Sq. No. 10, 410087 Oradea, Romania; alonvigdorovits@gmail.com (A.V.); drovipop@gmail.com (O.P.); 2Center for Research and Innovation in Personalized Medicine of Respiratory Diseases, “Victor Babes” University of Medicine and Pharmacy, 300041 Timisoara, Romania; 3Bihor County Clinical Emergency Hospital, Gheorghe Doja, Street No. 65, 410169 Oradea, Romania; maria.koteles@yahoo.com; 4Faculty of Pharmacy, “Victor Babes” University of Medicine and Pharmacy, Timisoara Eftimie Murgu Sq. No. 2, 300041 Timisoara, Romania; 5Research Center for Pharmaco-Toxicological Evaluations, Faculty of Pharmacy, “Victor Babes” University of Medicine and Pharmacy, Timisoara Eftimie Murgu Sq. No. 2, 300041 Timisoara, Romania

**Keywords:** artificial intelligence, cancer patients, pathology, digitalization, oncology, telepathology, medical systems

## Abstract

**Simple Summary:**

The quality of care available to cancer patients is highly constrained by the presence of comprehensive pathology services. However, pathology infrastructure is severely lacking in most resource-scarce medical systems. In recent years, progress made in the digitalization of pathology workflows and the advent of clinical-grade AI tools for pathology have offered pathologists new ways to increase their efficiency and offer better care to more patients. In this review, we aim to provide an overview of the current state of digitalization and AI in pathology and how it impacts the practice of oncology. We also identify the main challenges of implementing digital pathology and AI in resource-poor settings and point out the different approaches needed to tackle these challenges.

**Abstract:**

The application of artificial intelligence to improve the access of cancer patients to high-quality medical care is one of the goals of modern medicine. Pathology constitutes the foundation of modern oncologic treatment, and its role has expanded far beyond diagnosis into predicting treatment response and overall survival. However, the funding of pathology is often an afterthought in resource-scarce medical systems. The increased digitalization of pathology has paved the way towards the potential use of artificial intelligence tools for improving pathologist efficiency and extracting more information from tissues. In this review, we provide an overview of the main research directions intersecting with artificial intelligence and pathology in relation to oncology, such as tumor classification, the prediction of molecular alterations, and biomarker quantification. We then discuss examples of tools that have matured into clinical products and gained regulatory approval for clinical use. Finally, we highlight the main hurdles that stand in the way of the digitalization of pathology and the application of artificial intelligence in pathology while also discussing possible solutions.

## 1. Introduction

Using artificial intelligence (AI) to better understand the complexities of cancer, as well as to facilitate accurate diagnoses and effective treatments, is one of the most coveted goals in modern medicine. After several decades of “AI winter”, the last several years have seen increasing enthusiasm for the potential solutions that AI could offer patients [1,2]. Several innovations, such as the advent of deep learning (DL), the increased digitalization of patient data, and cloud computing, have swayed expert opinion and generated a greater level of openness towards to the adoption of AI [3]. 

Pathology is the cornerstone of modern oncology, and its role has expanded far beyond offering correct diagnoses. Pathology is playing an ever-increasing role in determining the treatment and prognoses of cancer patients [4]. Advances in whole-slide imaging (WSI) technology and computer hardware have increased the adoption of digital pathology and paved the way for the implementation of AI solutions [5]. These solutions can aid pathologists in tumor classification and tumor grading and can offer patients additional prognostic information derived from traditional histopathology, assisting the prediction of survival or molecular alterations [6]. Despite extensive research in this field, very few solutions have made their way into routine clinical practice due to financial constraints and various regulatory hurdles [7]. Furthermore, there is a lack of studies assessing the impact of AI in clinical settings [8]. When taking into consideration the lack of digitalization and infrastructure characteristic of resource-poor medical settings, the challenge of bridging the gap between research and patients seems even more daunting. Examples of important clinical applications that have been developed in the field can be seen in the grading of prostate cancer, the evaluation of biomarkers such as Ki-67, and the prediction of microsatellite instability in colorectal cancer (CRC) from routine H&E WSI [9,10,11].

The primary objective of this review is to provide a comprehensive overview of the current research landscape in the field of computational pathology, with a focus on applications with that could have an immediate impact on oncology patients. These applications are summarized in Appendix A. This paper also discusses the issues associated with the clinical implementation of AI and summarizes the most important applications that have gained regulatory approval and made their way to the clinic while also highlighting the problem of cost-efficiency. Finally, we also explore the implementation of AI in pathology in resource-poor settings, discussing challenges and potential avenues for increasing digitalization and the adoption of AI.

## 2. Artificial Intelligence in Pathology: Research 

### 2.1. AI Tools for Tumor Classification 

The classification of tumors is one of the main challenges pathologists face in routine practice. DL provides powerful algorithms for image classification, such as convolutional neural networks (CNNs) that have been widely used in classifying tumors [12]. A DL model trained on 579 whole-slide images from one academic hospital has been used to classify whole-slide images of transbronchial lung biopsies into adenocarcinomas, squamous cell carcinomas, small-cell lung carcinomas, and non-neoplastic diseases, achieving an area under the curve (AUC) of 0.94 to 0.99 in four independent test sets [13]. Another study reported similar results for a model trained on 741 whole-slide images and tested on independent datasets collected from multiple centers (AUCs of 0.918–0.978) in a six-category classification task that included the main types of lung cancer, as well as tuberculosis, organizing pneumonia and normal lung disease [14]. In gastrointestinal pathology, researchers have used CNNs and recurrent neural networks (RNNs) trained on 4128 gastric cancer whole-slide images and 4036 colon cancer whole-slide images to classify biopsies into adenocarcinomas, adenomas, and non-neoplastic diseases in the stomach and colon, with AUCs ranging from 0.96 to 0.99 [15]. A CNN model trained on 1865 images from 703 patients performed a six-class classification task for both CRC adenocarcinomas and the main types of benign colonic polyps, with a mean diagnostic accuracy of 97.3% [16]. Applications in prostate cancer grading have been highly successful, with a study in The Lancet Oncology reporting a model trained on 5759 biopsies with an AUC of 0.99 for classifying benign versus malignant lesions, 0.978 in identifying a grade group of 2 or more, and 0.974 in identifying a grade group of 3 or more while also scoring higher than a panel of 15 pathologists, outperforming 10 of them [9]. A paper from the same group showed that a panel of 14 pathologists had greater agreement with an expert reference standard when performing Gleason grading with the help of AI on a dataset of 160 biopsies (quadratically weighted Cohen’s kappa 0.799 vs. 0.872; *p* = 0.019) [17]. A DL model trained on 32,537 whole-slide images has been used to predict the origin of cancers of unknown primaries, achieving concordance in 61% of cases and a top-three agreement of 82% [18].

### 2.2. AI Tools for Biomarker Quantification

The quantification of biomarkers has an increasingly important role in the diagnosis, prognosis, and treatment of cancer patients. However, the process of quantification is plagued by high interobserver variability, which has been documented for numerous biomarkers [19,20,21]. A HER2 scoring algorithm that performed cell identification via the heatmap regression of a CNN combined with augmented reality microscopy was used to improve the accuracy of HER2 interpretation (0.93 vs. 0.80) and improve consistency for HER2 1+ cases [22]. A contest for HER2 scoring using AI was started to provide a benchmark for comparing the performance of various algorithms [23]. One study developed an DL-assisted method for PD-L1 scoring in breast cancer using 100 tumor resection samples, and the method improved concordance across the evaluating pathologists from moderate (0.674) to excellent (0.950) [24]. In the context of non-small-cell lung cancer (NSCLC), one study used QuPath, a free open-source tool, in order to automate PD-L1 scoring by training a random tree classifier using 67 different morphological features [25]. The reported concordance correlation coefficient between the automated and manual scores was 0.925. Automated scoring showed the same level of concordance with the average of three pathologists’ scores as an individual pathologist at both the 1% and 50% thresholds [26]. Another study developed a deep learning architecture with three modules—tumor segmentation (Res50-UNet), nuclei detection (MicroNet), and positive membrane detection (PMD)—that predicted the tumor proportion score (TPS) with a high correlation with a subspecialist pathologist (mean absolute error of 8.65 and Pearson correlation coefficient of 0.9436). The method outperformed generalist and trainee pathologists [27]. In another study, the AI-based evaluation of Ki-67 in breast cancer was performed using a workflow based on the Inception v3 and ResNet architectures, and the findings were compared with the Ki-67 standard reference card assessment, with the authors reporting an intra-group correlation coefficient larger than 0.905 among the pathologists that used these two methods [28]. Another study used a deep learning model based on DenseNet in concert with fuzzy-set interpretations for hot-spot identification to assess the Ki-67 proliferation index in in situ ductal breast carcinomas and reported a mean absolute error of 0.024 [29]. 

### 2.3. AI Tools for Survival Prediction

The estimation of survival time is of vital importance in patient management. The semi-parametric Cox proportional hazards model is the preferred method for modeling patient survival. This approach is based on a linear model, which makes it unable to capture non-linearities between independent variables and the risk of death. DL models provide a way of capturing non-linear relationships by using non-linear activation functions; therefore, they can refine the Cox model [30,31]. One study approached this problem by extracting tumor patches (images of 512 × 512 pixels) from whole-slide images of bladder urothelial carcinomas, invasive breast carcinomas, and glioblastomas using a pre-trained tumor classifier and applying graph-based CNNs to aggregate the information across patches and predict survival. The mean absolute errors (MAEs) of the model were 123.2, 167.5, and 303.3 for bladder carcinomas, breast carcinomas, and glioblastomas, respectively [32]. Another paper explored the use of adaptive sampling to extract patches from whole-slide images. The patches were later clustered, and a DL model was trained on these clusters to make patient-level predictions, with C-index values ranging from 0.510 to 0.703 [33]. MultiSurv is a model architecture that was developed to take advantage of multimodal data, including WSI patches, clinical data, copy number variation, gene and miRNA expression, and DNA methylation. This model was applied to 33 different cancer types, producing non-proportional estimates of survival, with time-dependent C-indexes ranging from 0.554 to 0.988 [34].

### 2.4. AI Tools for Predicting Molecular Alterations

With the advent of the genomic era in medicine, molecular pathology has occupied an ever-increasing role in the management of cancer patients. However, given the expertise and equipment needed to run molecular assays, accessibility remains an issue. Multiple research efforts have explored whether AI could help predict the presence or absence of clinically relevant mutations from H&E-stained whole-slide images (Figure 1). 

A paper by Coudray et al. in 2018 showed that a deep CNN based on the Inception v3 architecture and trained on 1.634 WSI from TCGA can predict sx common mutations in NSCLC from whole-slide images, including EGFR, FAT1, SETBP1, KRAS, and TP53, with AUCs ranging from 0.733 to 0.856 [35]. A transformer-based pipeline was trained and evaluated on a multicenter cohort of over 13,000 patients for the end-to-end prediction of microsatellite instability on CRC surgical resection specimens, achieving a sensitivity of 0.99 and a negative predictive value of over 0.99 [36]. An interpretable attention-based DL model trained on images from 1240 patients was used to predict the major molecular subgroups in endometrial cancer, with AUROCs of 0.849 for POLE ultra-mutation, 0.844 for DNA mismatch repair (MMR) deficiency, 0.928 for p53-abnormal, and 0.883 for no specific molecular profile (NSMP) [37]. One study attempted to predict the ten most common mutations in hepatocellular carcinoma (HCC) using a deep learning model based on the Inception v3 architecture and trained on 491 whole-slide images. The model predicted four of them (CTNNB1, FMN2, TP53, and ZFX4), with AUCs on external cohorts ranging from 0.71 to 0.89 [38]. Another study developed a ResNet-18 based on a model trained on images from 307 patients in order to predict IDH mutation status, an important prognostic biomarker in low-grade gliomas, with an AUC of 0.667 [39]. Hierarchical deep multi-instance learning was used to predict mutations of five clinically relevant genes in the TCGA bladder cancer dataset (ATM, PIK3CA, ERBB2, FGFR3, ERCC2), with AUCs above 0.83 [40]. Regarding breast cancer, a study showed that a DL model based on the ResNet-101 architecture and trained on 461 whole-slide images can be used to predict mutations in TP53, RB1, CDH1, NF1, and NOTCH2, with AUCs of 0.729 to 0.852 [41], while another study managed to predict germline BRCA mutations from whole-slide images with an AUC of 0.766 using a ResNet architecture [42]. One group reported somatic BRCA mutation prediction in high-grade ovarian cancer with an AUC of 0.681 using a ResNet-based architecture trained on images from 867 patients [43]. In an effort to help differentiate noninvasive follicular thyroid neoplasms with papillary-like nuclear features (NIFTP) from other neoplasms, an Xception-based DL model was trained on 115 whole-slide images to predict the BRAF-RAS score. This predicted score had an AUC of 0.99 and 0.97 when restricted to follicular pattern neoplasms [44]. Another study attempted to predict BRAF mutation in melanoma by using deep learning alone (AUC = 0.71) and by combining deep learning (Inception v3 architecture) with clinical information and pathomics, achieving an AUC of 0.89 [45]. Lung adenocarcinoma whole-slide images were used to whether the predict tumor mutational burden (TMB) was high or low via a model based on the Inception v3 architecture and trained on images from 499 patients, achieving an AUC of 0.92 on a held-out dataset [46].

### 2.5. Generative AI and Synthetic Data

Generating ground truth for the development of AI algorithms is both expensive and time-consuming. Generative adversarial networks (GANs) could provide models with labeled synthetic data, bypassing these issues [47]. PathologyGAN is an approach that uses a relativistic average discriminator that generates high-quality images of breast cancer and CRC, achieving a Frechet Inception Distance (FID) of 16.65 (breast cancer) and 32.05 (CRC). Ratings from two expert pathologists found no significant differences between the generated images and real ones, [48]. Wei et al. used CycleGANs to generate synthetic images of colorectal polyps which at least two out of four gastrointestinal pathologists could not discriminate from real images during a Turing test while also using the synthetic images to improve a ResNet classifier AUC by over 10% [49]. Another potential application of GANs is the virtual staining of histological slides, which could reduce costs as well as variability in staining [50].

## 3. Artificial Intelligence in Pathology: Clinical-Grade Tools

### 3.1. Implementation of AI in Routine Practice

Despite the large amount of published research on the applications of AI in pathology, a very small number of research projects go on to become mature commercial products that can be used in routine clinical practice. Research generates prototypes that must be further developed, validated on multiple independent datasets, and integrated into existing clinical laboratory workflows [51]. Clinical AI algorithms are considered medical devices, therefore necessitating regulatory approval before being made commercially available.

One major initial technical challenge is developing high-quality software that is easy to maintain and update. The computer code used in research projects is usually poorly structured and error-prone. Software design principles like object orientation and rigorous testing must be implemented to ensure code that is stable and works properly [52]. The level of maturity of AI solutions can be evaluated using technology readiness levels (TRLs) [53]. Homeyer et al. adapted TRLs for AI solutions in pathology. There are a total of nine levels, with higher ones indicating more mature products (Figure 2). A research prototype which is only evaluated on a limited dataset in a peer-reviewed publication is level 3, a clinical prototype that has been evaluated in routine pathology practice is level 6, while a solution that has cleared regulation and has achieved the status of commercial product is level 9 [54]. The validation of AI algorithms requires multiple datasets that are representative of the target population to avoid biases [55]. Moreover, obtaining regulatory approval is challenging due to the lack of transparency and model explainability, as well as adaptive training, which can cause changes in an AI solution after it gains regulatory clearance. The final step is reimbursement, which can only be obtained after the solution’s clinical utility has been demonstrated.

### 3.2. Examples of AI Tools Approved for Clinical Use in the USA and/or EU

The Galen platform, developed by Ibex Medical Analytics, represents a suite of AI solutions used to facilitate diagnosis across multiple organs and pathologies. The first of these solutions is Galen Breast, currently CE-marked under the In Vitro Diagnostic Directive (IVDD) in the EU and registered with the UK Medicines and Healthcare products Regulatory Agency (MHRA), as well as with the Brazilian Health Regulatory Agency (ANVISA). The validation and real-world clinical application study of the solution was published in December 2022. The study showed that Galen Breast can detect invasive carcinoma with an AUC of 0.99 (specificity of 93.57% and sensitivity of 95.51%) and DCIS with an AUC of 0.98 (specificity of 93.79% and sensitivity of 93.20%). The algorithm can also differentiate between invasive ductal carcinoma and invasive lobular carcinoma with an AUC of 0.97 and make the distinction between high-grade DCIS and low-grade DCIS/atypical ductal hyperplasia with an AUC of 0.92. Tumor-infiltrating lymphocytes (TILs) were also identified with an AUC of 0.965 [56]. Galen Prostate is CE-marked under IVDD, as well as under IVDR, and registered with the UK MHRA and ANVISA. The validation study for this tool was published in August 2020. The model detected cancer with an AUC of 0.991 in the external validation set. It differentiated between low-grade (Gleason score 6 or atypical small acinar proliferation) and high-grade (Gleason score 7–10) with an AUC of 0.941. Gleason pattern 5 was detected with an AUC of 0.971, and the cancer percentage calculated by the pathologists in the study, and the algorithm showed good correlation (r = 0.882, *p* < 0.0001), with a mean bias of −4.14%. The algorithm also achieved an AUC of 0.957 for the detection of perineural invasion [57]. Galen Gastric is CE-marked under IVDD, with a validation study presented as a poster at USCAP 2022. On real-world data, the solution demonstrated an AUC of 0.945 for detecting gastric adenocarcinoma and high-grade dysplasia and an AUC of 0.966 for the detection of *Helicobacter pylori* [58]. 

HALO Prostate AI, developed by Indica Labs is a DL-based screening tool currently CE-marked under IVDD, and its validation study was published in August 2023. The model identified biopsies that contained tumors with sensitivity, specificity, and negative predictive values ranging from 0.971 to 1.000, 0.875 to 0.976, and 0.988 to 1.000, respectively, across multiple test cohorts [59]. Histotype Px, by DoMore Diagnostics (CE-marked under IVDD), uses deep learning to provide risk stratification for patients with CRC using WSI. The validation study of the product was published in September 2022. This method provided hazard ratios of 10.71 for high-risk versus low-risk patients and 3.06 for intermediate- versus low-risk patients. Estimates of 3-year cancer-specific survival were 97.2% for the low-risk group, 94.8% for the intermediate-risk group, and 77.6% for the high-risk group. Risk stratification based on current guidelines was less prognostic, with the low-risk group being represented by only 13% of the validation cohort versus 41% in the DL based risk stratification [60]. MSIntuit, produced by Owkin and CE-marked under IVDD, is a tool that uses DL to identify MSI status from H&E WSI, and it achieved a sensitivity and specificity of 97% in a blinded validation study, with excellent agreement on two different scanners (Cohen’s kappa: 0.82) [61]. 

Paige’s Prostate Suite applications are CE-marked under IVDD, and Paige Prostate Detect is also the only FDA-approved AI solution for pathology. In one validation study, Paige Prostate Detect was reported to have a sensitivity of 97.4% and a specificity of 94.8% at the WSI level, with an AUC of 0.99. Pathologist sensitivity and specificity in diagnosis with and without Paige Prostate were measured, with an increase in accuracy from 88.7% unassisted to 96.6% when assisted, reducing detection errors by 70% [62]. Another study reported that pathologists diagnosed atypical small acinar proliferation about 30% less often while also requesting fewer IHC studies (around 20% less) and fewer second opinions (around 40% less) when assisted by Paige Prostate. The median time required to read and report slides was about 20% lower when assisted [63].

Visiopharm offers EU IVDR-certified tools for breast cancer, including the IHC quantification of Ki-67, estrogen receptor (ER), progesterone receptor (PR), and HER2, as well as the detection of tumors in pan cytokeratin-stained slides, hot spot detection, invasive tumor detection, and metastasis detection [64,65]. Aiforia has developed multiple AI models that are CE-marked under IVDD for breast cancer (ER, PR, Ki-67 scoring), prostate cancer (Gleason grading), and lung cancer (PD-L1 scoring) [66]. Mindpeak offers a suite of tools that have received CE marking under IVDD for lung cancer (PD-L1 scoring) and breast cancer (ER, PR, Ki-67, HER2 scoring) [10]. Deep Bio has developed a CE-marked under IVDD deep learning tool for prostate cancer grading, showing high concordance between the model and the reference standard (quadratic-weighted Cohen’s kappa of 0.907) [67].

### 3.3. The Problem of Cost-Efficiency

One of the main advantages invoked for digital pathology and AI implementation is increased efficiency, particularly in fully digital workflows. One study calculated projected cost savings in the first 5 years after the implementation of a digital pathology system. Based on potential improvements in productivity and laboratory consolidation, the potential savings were estimated to be USD 12.4 million in an institution with 219,000 annual accessions [68]. Another cost–benefit analysis was performed for a department of 45 full-time consultant pathologists, 80,000 specimens per year, and a yearly budget of GDP 9 million. The results showed that an increase in productivity of 10% would allow the institution to break even in 2 years, while an increase of 15% would allow the institution to break even after 1 year. A pathology laboratory with half the volume would make a profit in 4 years after a productivity improvement of 10% [69]. An analysis from the Memorial Sloan Kettering Cancer Center, published in 2019, reported that digital pathology implementation could lead to a 93% reduction in the requests for physical slides. In addition, the authors estimated yearly IHC savings of USD 114,000 due to fewer ancillary tests being ordered. Overall cost savings were estimated at USD 267,000 per year. Factoring in digital workflow setup and maintenance costs, it was estimated that the breakeven point would be 7 years after the initial transition [70]. A Delphi study on the role of computational pathology in 2030 revealed strong agreement among experts regarding the fact that the regulatory requirements will generate important costs in the future because of the need for large-scale prospective studies. No consensus was reached concerning a potential decrease in cost-per-case and second-opinion consultations. Most experts predicted that cost-per-case would not decrease in the following 8–10 years [71]. Another study evaluated the financial impact of implementing an AI-based evaluation of MSI status in a hypothetical population of 32,549 patients receiving first-line treatment for newly diagnosed metastatic CRC. Eight different testing strategies were compared: (1) next-generation sequencing (NGS), (2) high-sensitivity polymerase chain (PCR) reaction or IHC panel, (3) high-specificity IHC panel, (4) high-specificity AI, (5) high-sensitivity AI followed by NGS, (6) high-specificity AI followed by NGS, (7) high-sensitivity AI and high-sensitivity IHC panel, and (8) high-sensitivity AI and high-specificity panel. High-sensitivity AI followed by confirmatory high-specificity PCR or IHC panel for the patients that test negative using AI proved to be the strategy that resulted in the largest savings (USD 400 million, 12.9%) when compared to NGS alone [72].

## 4. Digitalization and AI in Countries with Scarce Resources

### 4.1. The Challenge of Digitalization

The process of digitalization represents the first step needed to implement AI in pathology. Digital pathology is being progressively deployed in more and more departments across the world, with several papers reporting the experience of both public and private pathology laboratories [73,74,75,76]. However, securing funding for digitalization is difficult. The large initial investments associated with WSI scanners, associated IT infrastructure, and the storage requirements needed for the large sizes of whole-slide images (up to several Gb per image) are important hurdles, particularly in low-resource settings, where the funding of pathology departments is lacking. 

Digitalization generally follows one of two implementation models: a “hybrid” approach where the digital workflow is gradually introduced [74] and a second model that consists of adopting a fully digital workflow very shortly after implementation [77]. The main advantage of the “hybrid” model is that it allows pathologists to gradually tackle the learning curve of working with whole-slide images while still having access to glass slides. The fully digital workflow has the major advantage of saving the time of histotechnicians that would otherwise be spent on slide sorting and distribution while also allowing pathologists to benefit immediately from various digital pathology tools [77].

Before implementing the digital workflow, it is necessary to have a dedicated information technology infrastructure in place. This includes a broadband internet connection, a sample tracking system integrated with the laboratory information system (LIS), a WSI scanner, and pathologists’ workstations [78]. Implementing the required infrastructure, together with subsequent quality-control tests and validation, can take up to several months [74,76]. The adoption of digital pathology allows for easier sharing of cases, enabling telepathology and remote consultations (Figure 3). Furthermore, it provides improved efficiency and productivity, as well as a smaller rate of interpretive errors [68]. Integration with molecular pathology and genomic research databases could further facilitate the use of digital pathology in clinical practice and research [75].

### 4.2. The Challenges of Pathology and Digitalization in Developing Countries 

The increasing burden of cancer in developing countries is creating new challenges given the lack of economic and healthcare infrastructure. The lack of proper diagnoses and treatments decreases the effectiveness of cancer control, which is based on surveillance and screening programs. These approaches only benefit patients if there is access to appropriate patient management once an early diagnosis of cancer has been made [79]. It has been found that the human development index has a significant negative correlation with the cancer mortality/incidence ratio [80].

As a critical first step in the management of cancer patients, pathology services are also severely underdeveloped in many parts of the world. Many of the scarce resources available are invested in offering treatment rather than diagnostic services [81]. According to a report from 2023, there are over 108,000 active pathologists in 162 countries and territories, corresponding to an average of 14 pathologists per million inhabitants. The disparities are glaring, with the United States having access to 65 pathologists per million, while African countries, on average, have less than 3 pathologists per million. Over two-thirds of the global pathologist workforce resides in 10 countries [82]. Other challenges faced by pathology as a speciality in developing countries include the variable standards of training, long turnaround times, lack of quality synoptic reporting, and the lack of visibility of pathology as a key component in the multidisciplinary clinical team [81]. Even though digitalization could increase accessibility to quality pathology services via telepathology or increase productivity in areas with insufficient pathologists, it requires a significant initial investment before benefits can be reaped. This might deter pathology laboratories in countries with scarce resources from making the transition to a digital workflow [74].

### 4.3. Telepathology as a Stepping Stone for Digitalization in Developing Countries

Telepathology is considered one of the main tools for overcoming pathologist shortages, with several successful examples of implementation being present in the literature [83,84,85]. The are four main approaches to telepathology: static images, whole-slide image scanning, dynamic nonrobotic telemicroscopy and dynamic robotic telemicroscopy. The static image approach requires minimal technical infrastructure but necessitates an appropriate selection of relevant diagnostic fields. Following the training period, static image telepathology diagnoses showed a concordance of 97% with a subsequent review of slides. WSI does not require the selection of diagnostic fields and allows pathologists to examine entire specimens at different magnifications. However, the costs of acquiring slide scanners and setting up the appropriate IT infrastructure and support prove quite prohibitive in low-resource environments. Dynamic nonrobotic telemicroscopy represents the video transmission of a microscope image in real-time across the internet, with a pathologist “driving” the microscope. This method is dependent on the quality of the internet connection and on image resolution but provides whole-slide imaging at a much lower cost. Dynamic robotic telemicroscopy allows a pathologist to control the microscope remotely. This approach is used less frequently due to costs and technical issues [86].

### 4.4. The Potential Uses of Social Media

Social media could help bridge knowledge gaps among pathologists and foster an inclusive environment for pathologists from developing countries. One retrospective survey-based study aimed to assess whether medical students that used Twitter during their pathology rotation found that it facilitated learning. Most respondents thought that Twitter (currently X) helped them retain more information while also being useful for professional development [87]. Twitter has also been described as a potential tool for case consultation and research collaborations, especially for pathologists in the developing world, who have less well-developed professional networks [88]. An article addressing the ethical issues of pathology and social media highlighted the fact that there are no significant differences regarding patient privacy when images are shared on social media platforms compared to image publication in a medical journal and that, with responsible use, the benefits far outweigh the risks [89]. 

Recently, given the growing number of pathologists creating content, social media has also proved to be a useful resource for the training of AI models. A group of researchers used pathology images from tweets, as well as images from published literature, to develop multimodal DL models that identify histopathology stains, classify tissues, and differentiate disease states with AUROCs of 0.805 to 0.996. Based on these models, the researchers created a social media bot that can help pathologists obtain real-time feedback on the cases that they post [90]. A recent study developed a method called pathology language-image pretraining (PLIP), which is a multimodal AI model trained on OpenPath, a dataset of pathology images curated from Twitter hashtags paired with natural language descriptions and pathology data from other websites. PLIP was then tested on four external datasets and achieved state-of-the-art results for zero-shot classification tasks, with F1 scores of 0.565–0.832. PLIP also allows users to find similar cases using image or natural language searches [91].

## 5. Conclusions and Future Directions

In this review, we provided an overview of the main research directions in computational pathology applied to oncology. We then discussed the challenges associated with bridging the gap from research to the clinic and highlighted the most important tools that have made their way through regulatory hurdles and are available for clinical use. We also discussed the avenues by which digital pathology and AI can be implemented in low-resource settings while also considering the existing limitations. 

A major issue identified was the striking discrepancy between the burgeoning research landscape of computational pathology and the scarcity of tools available for clinical use by pathologists. Most research projects produce intriguing prototypes that must be further developed to become clinical-grade tools. This development process requires making significant investments and passing ever-increasing regulatory hurdles, which discourages public sector involvement. In the future, more public and non-profit interest in computational pathology could greatly increase access to pathology services in developing countries and lower resource settings.

In conclusion, the potential of AI in the field of pathology has yet to be fully realized. The lack of digitalization of pathology services in developing countries generates difficulties when it comes to being involved in the creation of AI tools. It is essential for researchers, clinicians, companies, and various other stakeholders to collaborate to overcome future challenges and offer cancer patients equitable access to AI-enhanced pathology services.

## Figures and Tables

**Figure 1 cancers-15-05692-f001:**
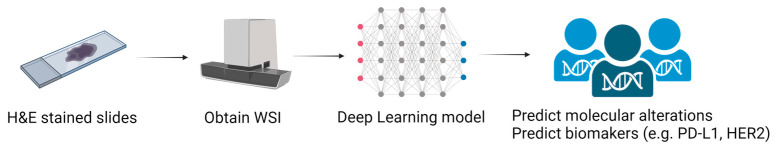
Predicting molecular alterations from H&E-stained whole-slide images.

**Figure 2 cancers-15-05692-f002:**
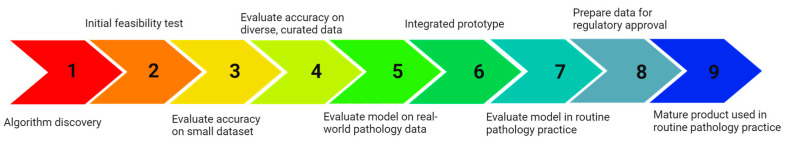
Technology readiness levels in pathology.

**Figure 3 cancers-15-05692-f003:**
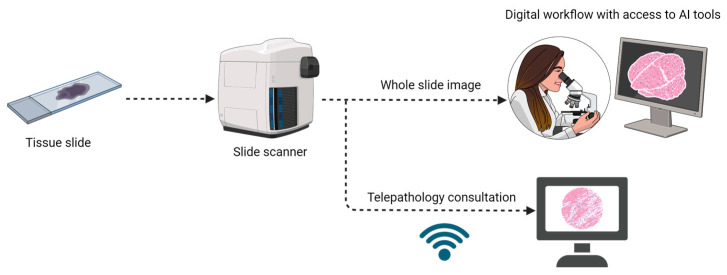
Digital pathology workflow.

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
