# Peer review of "Breaking Barriers: AI’s Influence on Pathology and Oncology in Resource-Scarce Medical Systems"

_cancers, 2023, doi:10.3390/cancers15235692_

Round 1

Reviewer 1 Report

Comments and Suggestions for Authors

This paper was a review article focusing on research in computational pathology, with focus on oncology.

1.     In the section for AI tools for tumor classification (Section 2.1), the results mentioned in the review is good, but can you elaborate on the dataset, specifically the number of subjects and where was this data obtained (in-house / public or external hospital).

2.     In the section for AI tools for biomarker quantification (Section 2.2), the there are few results mentioned for biomarker quantification, it would be useful to mention the AI models, architecture and method of classification for the results mentioned in this section.

3.     In the section for AI tools for Survival Prediction (Section 2.3), the results for the papers are not mentioned. Can you list the results for all the papers cited in this section, specifically for the survival prediction task.

4.     In the section for AI tools for Predicting molecular alterations (Section 2.4), since it is a review article, some of the models used in the paper were mentioned, it would be a good idea to be consistent and mention the data and models used in all papers mentioned in this section.

5.     In the section for AI tools for Generative AI and synthetic data (Section 2.5), the papers have been cited without any results or data statistics, it would be useful to have this values for the review.

6.     In the section for The challenges of Digitalization, since the authors have mentioned 2 different approaches of digitalization, hybrid and fully digital model, it would be helpful if there was a cost analysis comparison for both the models with the traditional model.

Author Response

Dear Reviewer,
Thank you for your insightful comments and suggestions. We appreciate the time and effort you have put into reviewing our manuscript. We have carefully considered each point and have made revisions accordingly. Please find our responses to your comments below:

1. We have elaborated on the datasets. Specifically the number of cases in Section 2.1.

2. The AI models used in Section 2.2 are now mentioned.

3. For section 2.3, the results of the survival analysis papers are now included.

4. In section 2.4, we have elaborated on the models used.

5. For section 2.5, the metrics used to evaluate the Generative AI models are now mentioned.

6. Unfortunately, cost analysis studies in digital pathology implementation are very few and we could not find such a comparison in the literature. However, this is something we will pursue in our future studies. We greatly appreciate this insightful idea. 

We hope that these revisions address your concerns and improve the quality of our manuscript. We look forward to your feedback on these changes.
Best regards,
The authors.

Reviewer 2 Report

Comments and Suggestions for Authors

In recent years, the field of Artificial Intelligence (AI) in oncology has grown exponentially. AI solutions have been developed to tackle a variety of cancer‐related challenges. AI in oncology has demonstrated accurate technical performance in image analysis, predictive analytics, and precision oncology delivery. In this paper, the authors present the review of   the current state of digitalization and AI in pathology and how it impacts the practice of oncology.

Comments.

1.      A summary table may be included which should contain the pros and cons of all papers.  

Comments on the Quality of English Language

Nil

Author Response

Dear Reviewer,
We appreciate the time and effort you have put into reviewing our manuscript. We have carefully considered your suggestion accordingly and included a table that contains all of the research applications mentioned in our manuscript.
Best regards,
The authors.

Reviewer 3 Report

Comments and Suggestions for Authors

This review revises AI in pathology field. The manuscript is well written, it is easy to read and to understand. I covers some of the most known applications of AI, but not all. The text may improve with summarize data in tables.

Specific comments:

(1) In section 2.2., AI-based analysis technique may in some situations be superior to pathologist. However, can AI identify the regions of interest to be analyzed? Is this part done by the pathologists?

(2) Line 142, regarding "DL models provide a way of capturing non-linear relationships and can therefore refine the Cox mode". Please explain how.

(3) Section 2.3. What do you mean by "patches"?

(4) In section 2.4, are all the examples about predicting molecular alterations based solely on the histological features evaluated with deep learning?

(5) Regarding section 2.5. What database of images were the source of the generative synthetic data?

(6) Line 231, regarding "lack of transparency and model explainability". Please be aware that, for example, neural networks are not a "black box", the code and the "insights" can be check in the synaptic weights.

Synaptic weights: displays the coefficient estimates that show the relationship between the units in a given layer to the units in the following layer. The synaptic weights are based on the training sample even if the active dataset is partitioned into training, testing, and holdout data. Note that the number of synaptic weights can become rather large and that these weights are generally not used for interpreting network results.

(7) Section 4.1. Please do not forget the cost of storage of digital images in servers. A whole-tissue section of average size (3x2cm), when digitalized at 400x resolution has a size of around 4-5 Gb. This means that the necessary resources for storage of all slides is huge. To date, no solution is found. Usually, histological slides are kept for 10 years, while FFPET blocks are kept forever. Scanner, and workstations, and screens are the easy part. You may mention the LTO technology of Fujifilm (tape).

(8)  You may make a table showing the most relevant AI applications in pathology, with the reference of the manuscript.

(9) The title is  "Breaking Barriers: AI’s Influence on Pathology and Oncology in Resource-Scarce Medical Systems". With the provided information of this review, I do not think AI in influencing the resource-scare medical systems. On the contrary, low income countries have few pathologist, less technical resources, and no AI at all. In future, AI tools will be available to rich countries, not poor. You may change the title.

(10) May may comment and cite about immune oncology.

Cancers 2022, 14(21), 5318; https://doi.org/10.3390/cancers14215318 BioMedInformatics 2021, 1(1), 18-46; https://doi.org/10.3390/biomedinformatics1010003 Cancers 2021, 13(24), 6384; https://doi.org/10.3390/cancers13246384 

García R, Hussain A, Chen W, Wilson K, Koduru P. An artificial intelligence system applied to recurrent cytogenetic aberrations and genetic progression scores predicts MYC rearrangements in large B-cell lymphoma. EJHaem. 2022;3(3):707-721. Published 2022 May 16. doi:10.1002/jha2.451

Buciński A, Marszałł MP, Krysiński J, Lemieszek A, Załuski J. Contribution of artificial intelligence to the knowledge of prognostic factors in Hodgkin's lymphoma. Eur J Cancer Prev. 2010;19(4):308-312. doi:10.1097/CEJ.0b013e32833ad353

Gaur K, Jagtap MM. Role of Artificial Intelligence and Machine Learning in Prediction, Diagnosis, and Prognosis of Cancer. Cureus. 2022;14(11):e31008. Published 2022 Nov 2. doi:10.7759/cureus.31008

Shmatko A, Ghaffari Laleh N, Gerstung M, Kather JN. Artificial intelligence in histopathology: enhancing cancer research and clinical oncology. Nat Cancer. 2022;3(9):1026-1038. doi:10.1038/s43018-022-00436-4

He B, Bergenstråhle L, Stenbeck L, et al. Integrating spatial gene expression and breast tumour morphology via deep learning. Nat Biomed Eng. 2020;4(8):827-834. doi:10.1038/s41551-020-0578-x

Hashimoto N, Takagi Y, Masuda H, et al. Case-based similar image retrieval for weakly annotated large histopathological images of malignant lymphoma using deep metric learning. Med Image Anal. 2023;85:102752. doi:10.1016/j.media.2023.102752

Miyoshi H, Sato K, Kabeya Y, et al. Deep learning shows the capability of high-level computer-aided diagnosis in malignant lymphoma. Lab Invest. 2020;100(10):1300-1310. doi:10.1038/s41374-020-0442-3

Tomita H, Yamashiro T, Iida G, Tsubakimoto M, Mimura H, Murayama S. Unenhanced CT texture analysis with machine learning for differentiating between nasopharyngeal cancer and nasopharyngeal malignant lymphoma. Nagoya J Med Sci. 2021;83(1):135-149. doi:10.18999/nagjms.83.1.135

Maeda Y, Watanabe T, Izumi T, Kuriyama K, Ohno S, Fujimuro M. Biomolecular Fluorescence Complementation Profiling and Artificial Intelligence Structure Prediction of the Kaposi's Sarcoma-Associated Herpesvirus ORF18 and ORF30 Interaction. Int J Mol Sci. 2022;23(17):9647. Published 2022 Aug 25. doi:10.3390/ijms23179647

Bobée V, Drieux F, Marchand V, et al. Combining gene expression profiling and machine learning to diagnose B-cell non-Hodgkin lymphoma. Blood Cancer J. 2020;10(5):59. Published 2020 May 22. doi:10.1038/s41408-020-0322-5

Author Response

Dear Reviewer,
Thank you for your insightful comments and suggestions. We appreciate the time and effort you have put into reviewing our manuscript. We have carefully considered each point and have made revisions accordingly. Please find our responses to your comments below:

1. At least for biomarker quantification, the selection of regions of interest is done by the pathologist most of the time. Unsupervised learning could however (with enough training examples) be used to perform automated selection of regions of interest.

2. Deep learning models can capture nonlinearities by using non-linear activation functions such as ReLU.

3. Patches are square images of a certain size selected from the WSI. We have now added this clarification to the manuscript.

4. Predictions of these molecular alterations are attempted from routine H&E slides. We have now added this clarification in the manuscript text and figures.

5. Images used to train the generative models were taken from archives of several cancer institutes and hospitals in Germany (National Center for Tumor diseases), Netherlands (Netherlands Cancer Institute), and Canada (Vancouver General Hospital) which are publicly available.

6. While we agree that synaptic weights provide some information, the insights to be gained from this way of looking at model functions are limited. Even tools such as class activation maps which produce visual correlates that are hard to interpret by a pathologist and systematically deducing what the models are looking at are very difficult and inconsistent.

7. Storage of images is a major hurdle in the face of digitalization. We have added this observation to the manuscript.

8. We have included a table highlighting the research applications mentioned in our manuscript

9. While we agree that the impact of AI in pathology on a large scale will first be felt in developedcountries, we strongly believe that this revolution in pathology is a potential gap closer between medical systems with different amounts of resources. As we highlighted in our manuscript, telepathology is an important stepping stone that can increase access to both pathology expertise and AI tools, sometimes without the need to have an in-house IT infrastructure. Thus, we decided to keep our original title.

10. Due to the burgeoning nature of the field of immune-oncology and the potential impact of many other modalities of diagnosis other than pathology we have decided to not include this field in our review.

We hope that these revisions address your concerns and improve the quality of our manuscript. We look
forward to your feedback on these changes.
Best regards,
The authors.